# The Effects of Resistance Exercise on the Cardiorespiratory Tissue of Rats with Parkinson’s Disease

**DOI:** 10.3390/ijerph20042925

**Published:** 2023-02-08

**Authors:** Graziele Mayra Santos Moreira, Laila Cristina Moreira Damázio, Silvana Venâncio da Silva, Augusto Targino Silveira, Isabella Giordano Mesquita, Luana Aparecida de Sousa Silva, Luan Alves Pereira, Ana Clara Silva Costa, Ismael Augusto Lima Santos, Maria Eduarda Paiva Campos, Luna Sara Campos Vaz, Zenaide Alves Cardoso, José Victor Ribeiro Silva Gomes, Paulo Henrique Almeida Campos Júnior, Liliam Midori Ide

**Affiliations:** 1Graduate Program in Morphofunctional Sciences (PPGCM), Federal University of São João del-Rei (UFSJ), Sao Joao del Rei 36307-352, Brazil; 2Physiotherapy Department, Presidente Tancredo de Almeida Neves University Center (UNIPTAN/Afya), São João del Rei 36307-251, Brazil; 3Medicine and Biological Sciences Course, Dom Bosco Campus, Federal University of São João del-Rei (UFSJ), Sao Joao del Rei 36301-158, Brazil

**Keywords:** Parkinson’s disease, exercise, heart, muscles

## Abstract

Parkinson’s Disease (PD) affects the cardiorespiratory system, causing an increase in the left ventricular mass in the heart and respiratory muscle weakness in relation to healthy individuals. The objective of this study was to investigate the effects of progressive resistive physical exercise on the vertical ladder on the histomorphometry of cardiac tissue and respiratory muscles in rats with PD. Seventy male Wistar rats, aged 40 days old, were divided into Parkinson’s (PD) and Sham (SH) groups; they were also subdivided into groups that performed progressive resistive physical exercise on the vertical ladder Before Surgery (ExBS), After Surgery (ExAS), and Before and After Surgery (ExBAS). The physical training was carried out before and/or after PD induction. Exercise was performed five times a week for 25 min/day for four or eight weeks. PD induction was conducted via electrolytic stimulation of the *Substantia nigra* of the animals’ brains, adopting the following stereotaxic table coordinates: −4.9; lateral medium equal to 1.7, and dorsoventral equal to 8.1. For the morphometric analysis of the heart, the following variables were calculated: relative weight and diameter and thickness of the left ventricle. The diaphragm and the myocardial, intercostal, and abdominal muscles were stained with Hematoxylin and Eosin (HE). The histomorphometric analysis was performed using the ImageJ software to assess the cross-sectional area of muscles and the number of muscle fibers. Progressive resistance exercise promoted the hypertrophy of respiratory muscles and the left ventricle in animals with PD.

## 1. Introduction

Parkinson’s Disease (PD) is defined as a chronic and progressive neurodegenerative disorder involving the death of dopaminergic neurons. The decrease in dopamine production causes initially subtle motor alterations, with most patients being unable to identify the exact moment of symptom onset [1].

The cause of PD is believed to be related to defects in enzymes involved in the degradation of specific proteins, such as alphanucleic proteins and parkintin, contributing to the death of neurons in the *Substantia nigra*. In addition, there is evidence that the cause may be related to medication use, brain injuries, stress, and genetic factors [2,3].

Currently, PD is considered a global concern. In addition to generating physical dependence, there are indications of a significant increase in the number of cases in coming years [4]. The Brazilian Institute of Geography and Statistics (IBGE) estimates that 200,000 people have Parkinson’s in Brazil, and the World Health Organization (WHO) revealed that 1% of the global population over the age of sixty-five has the disease [5].

Complications resulting from PD can affect multiple body systems. Respiratory involvement is worrisome, as it is considered the leading cause of death. Individuals with the disease present chest rigidity, reduced chest expansion, complaints of dyspnea, difficulty coordinating upper airway muscles, and reduced respiratory muscle function and strength, which contribute to respiratory dysfunction and susceptibility to restrictive lung diseases. Thus, it is valid to hypothesize that respiratory mechanics may be impaired as a result of changes in muscle fibers and atrophy of respiratory muscles [6].

Furthermore, it has been pointed out that PD affects the cardiovascular system and promotes sympathetic dysfunction, with a loss of cardiac noradrenergic innervation. Some studies also describe dysautonomy, orthostatic hypotension, supine hypertension, increased risk of ischemic heart disease, and heart failure [7]. According to Flores et al. [8], individuals with PD show increased left ventricular mass indices compared to healthy individuals (114.2 ± 38.4 vs. 94.1 ± 26.4 g/m^2^), greater concentric remodeling, diastolic dysfunction, and less cardiac contractility, indicating that the greater the dysautonomy, the greater the ventricular hypertrophy.

In recent years, the applicability of high-intensity progressive resistance exercise has been tested in several diseases, raising controversies regarding its efficacy and safety in cardiovascular and cerebrovascular diseases [9,10]. There is a greater tendency to use moderate-intensity exercises in brain diseases, as is recommended in the latest guideline for people with PD [11]. In fact, no study has yet demonstrated the effects of high-intensity progressive resistance exercises on the cardiorespiratory system in animal and human models with PD.

Considering the issues mentioned above, which demonstrate the scarcity of studies that relate progressive resistance exercises of high intensity in PD as well as the absence of studies evaluating cardiorespiratory changes, the aim of the present study was to investigate the effects of a high-intensity progressive resistance physical exercise, performed on the vertical ladder, on the cardiorespiratory tissue of rats with electrolytic-stimulation-induced PD.

## 2. Materials and Methods

### 2.1. Study Animals

This study used 70 male Wistar rats (*Rattus norvegicus*, var. albinus), aged 40 days old and with a body weight of 200 g. The sample was distributed into two groups: the Parkinson’s Disease (PD) group, submitted to PD induction using the electrolytic lesion model of the *Substantia nigra* of the midbrain, and a Sham group (SH), without electrolytic stimulation. In addition, the animals were subdivided according to the experimental protocol adopted into sedentary (Sed), progressive resistance exercise before surgery (ExBS), progressive resistance exercise after surgery (ExAS), and progressive resistance exercise before and after surgery (ExBAS), as shown in Figure 1.

The research project was submitted and approved by the Ethics Committee for the Use of Animals (CEUA) under protocol No. 9049140321.

### 2.2. PD Induction

For PD induction, the animals were anesthetized intraperitoneally using a combination of Ketamine (75 mg/kg) and Xylazine (10 mg/kg) and positioned with their head fixed in a stereotaxic device. Here, asepsis of the cephalic region, trichotomy, and the removal of the subcutaneous tissue and periosteum for the visualization of the Lambda and Bregma areas were performed, followed by the insertion of electrodes for electrolytic stimulation in the *Substantia nigra* region at the following coordinates: −4.9; lateral medium equal to 1.7, and dorsoventral equal to 8.1 [12]. The electrodes caused injury to the *Substantia nigra* using a current load of 1.0 mA for 10 s. They remained in place after the injury for approximately 3 min [13]. Finally, the injured site was sutured with a surgical thread. A histomorphometric analysis of the *Substantia nigra* of the midbrain enabled us to assess the lesion area and confirm PD induction.

In the PD group, an electrolytic lesion was made in the *Substantia nigra* using a continuous current of 1.0 mA for 10 s, after which the electrode remained for 3 min and was later removed. The Sham groups were also submitted to the same procedure, however, without the application of the electric current. After procedure completion, the electrodes were removed and the region was sutured using surgical thread [12].

### 2.3. Physical Training

In order to perform the high-intensity progressive resistance physical exercises, a vertical ladder measuring 10 cm in length and 18 cm in width and with an inclination of 80° was used. There was also a housing box at the top end of the ladder, which measured 20 cm high, 20 cm wide, and 20 cm long. The animals were submitted to exercises before, after, or before and after PD induction.

All animals underwent an adaptation period on the vertical ladder for three days, in which they remained in the housing chamber for 60 s to familiarize themselves with the environment. Next, they were placed on the ladder at a distance of 35 cm from the housing chamber, then 55 cm, and, finally, 110 cm, thus performing three ascents per day without a load.

The rats performed progressive resistance physical exercises five times a week, with a training duration of 25 min per day for four weeks in groups ExBS and ExAS and four weeks before and four weeks after surgery in the ExBAS group. The sedentary group did not carry out any activity; therefore, it was not submitted to exercise on the vertical ladder. Throughout the entire experimental period, the sedentary subjects remained confined in a polyethylene box for four weeks before surgery and four weeks after surgery. They were only removed from the boxes for care, such as weighing and cleaning.

The progressive resistance physical exercise protocol with weights was adopted using loads of 50%, 75%, 90%, and 100% of the animal’s body weight. The weights were fixed to the proximal portion of the animals’ tails with wool yarn wrapped in adhesive tape. On each training day, the rats carried out eight sets of climbing (ascents): the first two with a load equal to 50% of the animal’s total body weight, the third and fourth with 75% of their body weight, the fifth and sixth with 90% of their body weight, and the final two ascents with 100% of the animal’s total body weight [14,15,16]. The maximum heart rate (HRmax) of the animals was monitored, in addition to their oxygen saturation level (SatO_2_), to verify that the exercise reached 80 to 95% of the animal’s HRmax, considered high intensity [17].

### 2.4. Histological Processing and Histomorphometric Analysis of the Substantia Nigra of the Rat Brain

After euthanasia, part of the right lung, heart, diaphragm muscle, intercostal muscle, and *Rectus abdominis* muscle were removed and placed in 10% formaldehyde for 24 h, followed by 70% alcohol. The brain was placed in 4% formaldehyde for 24 h, followed by 70% alcohol, and sliced into 1.0 mm sections in the coronal plane, removing slices of *Substantia nigra* from the midbrain.

The collected structures followed the processes of dehydration with alcohol (60%, 70%, 80%, 90%, and 100%), diaphanization in Xylol (1, 2, and 3), and were bathed twice in paraffin before being embedded in paraffin blocks. The paraffin blocks were cut with a 4 µm thick microtome. Finally, the sections were immersed in water at room temperature to fixate the paraffin tapes with the biological tissue using a glass slide, which was then placed in an oven at approximately 70–80 °C for 15 min.

A histochemical analysis of the *Substantia nigra* of the midbrain was performed by observing the neuronal cells and confirming their degeneration using the Nissl method, with the immersion of the slide in a cresyl violet solution to evidence the cytoplasm of the neurons and Nissl corpuscles.

The adopted protocol consisted of the following procedures: the slides were immersed for five minutes in flasks containing xylene 1 and xylene 2; 100% alcohol, 100% alcohol, 95% alcohol, and 70% alcohol. Subsequently, they were immersed in a 0.5% cresyl violet solution (neuron-specific marker) for 30 min, followed by dehydration for five min in flasks containing 70% alcohol, 95% alcohol, 100% alcohol, and 100% alcohol. Next, they were cleared in xylene 1 and xylene 2 and dried in ambient air for 24 h.

The slides were observed under a microscope (Motic) using a 4× lens, followed by image capture and the qualitative analysis of neuronal degeneration in the *substantia nigra* [18,19].

### 2.5. Histological Processing and Histomorphometric Analysis of the Parenchyma and Respiratory Muscle Tissue

Lung, parenchyma, and respiratory muscles were stained with hematoxylin and eosin (HE) [20] as follows: deparaffinization in Xylol 1 (15 min) and Xylol 2 and 3 (dipped 20 times), hydration in alcohol (60%, 70%, 80%, 90%, and 100%, sequentially), 20 times in each container, washed under running water for 3 min, stained with hematoxylin (3 min), washed under running water, stained with eosin (30 s), and washed under running water. Finally, the slides were dehydrated in alcohol (50%, 60%, 70%, 80%, 90%, and 100%, sequentially) fixated in xylene (1, 2, 3, 4, 5, and 6), and dipped 20 times in each container, followed by the application of acrylic varnish. They were then placed to dry in ambient air for 24 h.

The slides were photographed using the Motic Images Plus 2.0 imaging software and a digital camera (Moticam 580) attached to a microscope (new Optical Systems, 1801). The images were analyzed using the ImageJ Software (Image-Pro Plus, version 4.5, Windows 98) [20].

Lung tissue slides were analyzed at 100× magnification to measure alveoli diameter and thickness. The analysis of the cross-sectional area of the diaphragm and the intercostal and rectus abdominis muscles was performed at 400× magnification (area of 205,595.40 µm^2^).

### 2.6. Analysis of the Heart

The morphometric analysis of the heart was carried out using the calculation of relative weight (weight of the heart × 100 and divided by the weight of the animal) [21]. After sectioning the heart at the midpoint between the apex and the coronary sulcus, the organ was photographed with a cell phone (iPhone 11, Apple) to analyze the internal diameter and thickness of the left ventricle (LV) with the ImageJ software.

### 2.7. Heart Rate Analysis

Cardiac functions were evaluated daily with the aid of a multiparametric monitor (Codec), which was used to measure the animals’ heart rate (HR) before and after vertical climbing to verify the intensity of physical exercise.

### 2.8. Statistical Analysis

Data analysis was performed using the statistical program Graph Pad Prism, version 5.0. The following statistical tests were used: ANOVA—one-way and two-way to compare the means between groups—and Turkey’s post-hoc test to evaluate more than one variable. The results were expressed as means ± standard error of the means, and the significance level adopted was 5%.

## 3. Results

The PD-induction model adopted promoted electrolytic damage to the *Substantia nigra* of the midbrain in all animal groups submitted to the procedure (Figure 2).

The results showed that the lung tissue morphology was not altered in any of the groups; the alveoli count in the groups (Figure 3A, *p* = 0.6739) presented approximate mean values. The values related to the thickness of the alveolar walls also showed no difference among groups (Figure 3B, *p* = 0.3281).

The evaluation of the respiratory muscles revealed that the PD and SH groups had similar cross-sectional areas when performing the same training (*p* > 0.999), but that were different (*p* < 0.0001) when comparing the three forms of training (ExBS, ExAS, and ExBAS) with the sedentary groups.

The diaphragm muscle (Figure 4A) exhibited muscle hypertrophy, with the PD-ExBS group presenting a greater area in relation to groups PD-ExAS (*p* = 0.0463) and SH-ExAS (*p* = 0.007). The PD and SH-ExBAS animals presented a larger area (PD-ExBAS vs. SH-ExAS, *p* = 0.049; SH-ExBAS vs. PD-ExAS, *p* = 0.018; and SH-ExBAS vs. PD-ExAS, *p* = 0.0023).

The analysis of the intercostal muscle (Figure 4B) demonstrated that the PD and SH groups submitted to the PD-Ex protocol had greater cross-sectional areas when compared to the ExAS (PD-ExBAS vs. SH-ExAS, *p* = 0.0019; PD-ExBAS vs. PD-ExAS, *p* = 0.025) and (SH-ExBAS vs. PD-ExAS, *p* = 0.0041; SH-ExBAS vs. SH-ExAS, *p* = 0.0003).

In the rectus abdominis muscle, muscle hypertrophy occurred similarly in the three trained groups (ExBS, ExAS, and ExBAS), demonstrating differences when compared to the sedentary groups (*p* = 0.0001).

When analyzed at rest and after training, the heart rate showed similar values among the PD and SH groups (*p* = 0.056), regardless of the type of training performed (*p* = 0.57). There was an increase in heart rate with the performance of Ex (*p* < 0.0001), as shown in Figure 5.

The morphometric analysis of the LV showed that there was a significant difference between the ventricular diameters (Figure 6B,E) and the PD-ExBS, PD-ExAS, and PD-ExBAS groups compared to the sedentary PD and SH groups (PD-ExBS vs. PD-Sed, *p* = 0.006; PD-ExBS vs. SH-Sed, *p* = 0.023), (PD-ExAS vs. PD-Sed, *p* = 0.0032; PD-ExAS vs. SH-Sed, *p* = 0.0135), and (PD-ExBAS vs. PD-Sed, *p* = 0.0009; PD-ExBAS vs. SH-Sed, *p* = 0.0043). Regarding its thickness (Figure 6C,D), we noted that the PD-ExBAS and SH-ExBAS groups had a greater LV thickness when compared to the sedentary PD and SH groups (PD-ExBAS vs. SH-Sed, *p* = 0.026; SH-ExBAS vs. PD-Sed, *p* = 0.0020; and SH-ExBAS vs. SH-Sed, *p* = 0.0003).

## 4. Discussion

In the present study, there were no perceivable changes in the lung parenchyma of rats with PD compared to the SH groups. However, progressive resistance exercises of high intensity promoted hypertrophy of the respiratory muscles and myocardial hypertrophy when comparing the exercised animals to the sedentary ones. An increase in heart rate was also evidenced with the performance of the progressive resistance physical exercises when comparing the heart rate at rest and after training.

Neuronal degeneration in the mesencephalic *Substantia nigra* was observed using the electrolytic lesion model, confirming the induction of PD. The electrolytic lesion, despite differing from the lesion with the 6-OHDA toxin, is similar when using stereotaxic coordinates for induction. According to Gomes and Bel [13], there is a destruction of dopaminergic neurons in the *Substantia nigra* and the nigrostriatal pathway (caudate and putamen) in both models, confirmed by immunocytochemical and histochemical analyses with neurons positive for NOS and NADPH-d, respectively.

No significant difference was observed in the number of alveoli and the lung parenchyma of the animals in the PD and Sham groups. In humans, respiratory alterations are described in patients with PD and are the main cause of death in patients. Changes in posture, weakness, and muscle stiffness, which modify the biomechanics of the rib cage, promote deficits in lung expansion and hamper gas exchange, directly impacting respiratory function [22,23].

This study was the first to verify respiratory muscle hypertrophy with high-intensity progressive resistance exercise in PD, showing an increase in the cross-sectional area of the diaphragm and the intercostal and rectus abdominis muscles. It was also evidenced that hypertrophy occurred in the same way in rats in the PD and SH groups, demonstrating that the disease did not influence the increase in muscle mass.

Respiratory muscle training usually involves inspiratory resistive loading and, in humans, promotes an increase in the cross-sectional area and the proportion of muscle fibers (%) of the intercostal muscles upon biopsy. In rats, there is evidence of diaphragm muscle hypertrophy with an increased muscle area [24,25]. In the study by Bowen et al. [26], the repercussions of intense physical exercises on the respiratory muscles were evidenced; however, the authors analyzed obese rats with heart failure and did not find significant differences in the cross-sectional area and the number of muscle fibers when compared to the sedentary groups. In their study, high-intensity (90% of peak VO_2_) and moderate-intensity (60% of peak VO_2_) exercises were performed on a treadmill with an inclination of 25° for eight weeks.

During physical exercise, there is a linear increase in heart rate with increased exercise intensity. As expected, there was an increment in heart rate with the performance of progressive resistance physical exercises, and the heart rate at rest was higher in the PD-Sed group compared to the SH-Sed group. In addition to the increase in heart rate, studies show that reduced blood lactate and increased VO_2_max are common effects of high-intensity physical exercise in rats [9,27].

The heart analysis showed myocardial hypertrophy as an effect of progressive resistance exercise. Although no differences were found regarding cardiac weight and relative weight, significant macroscopic differences were evidenced, such as a smaller LV diameter in the PD-ExBS, PD-ExAS, and PD-ExBAS groups and a greater LV thickness (PD-ExBAS and SH-ExBAS) when compared to the sedentary PD and SH groups. Similar results were described in the study by Vasconcelos et al. [21], in which high-intensity physical exercise was performed on the vertical ladder in rats after cerebral ischemia. This outcome shows that the intense overload used in physical training produces concentric hypertrophy and can similarly favor animals with or without PD, as the increase in myocardial muscle mass promotes greater cardiac output and a decrease in resting HR and blood pressure [28].

Individuals with PD may present cardiac alterations, with greater concentric remodeling of the myocardium associated with a worsening of diastolic function and a greater chance of developing heart failure with initially preserved ejection fraction [8]. It is worrying that the structural alteration of the heart increases the chances of arrhythmias and unexpected death [29]. Thus, it was demonstrated that the effect of progressive resistance physical exercise triggers significant changes in the myocardium and physiological functions of the heart in animals with PD. These need to be further studied and clarified regarding the benefits and the ideal recommendations for these animals.

Despite the scarcity of studies demonstrating alterations in cardiorespiratory morphology after physical exercise, the importance of progressive resistance physical exercise in the cardiorespiratory system is noteworthy. However, tests that analyze cardiopulmonary function are still necessary to confirm the safety and effectiveness of the training, as well as to compare progressive resistance physical exercises such as climbing with other training modalities. Furthermore, the use of the maximum heart rate (HRmax) to establish exercise intensity in rats needs to be further evaluated, as other measures could be added to confirm exercise intensity, such as measurements of blood lactate and VO_2_max.

## 5. Conclusions

The high-intensity progressive resistance exercises on the vertical ladder did not promote morphological changes in lung tissue. However, they did trigger the hypertrophy of respiratory muscles, with an increase in the cross-sectional area of the diaphragm and the intercostal and rectus abdominis muscles. In the heart, the exercise caused a reduction in the diameter and an increase in the thickness of the left ventricle in animals with and without Parkinson’s disease.

Thus, it was possible to observe the benefits of progressive resistance exercise on the ladder for both the skeletal and cardiac muscle systems, proving the exercises to be an efficient therapeutic strategy for strengthening respiratory muscles and promoting myocardial hypertrophy.

## Figures and Tables

**Figure 1 ijerph-20-02925-f001:**
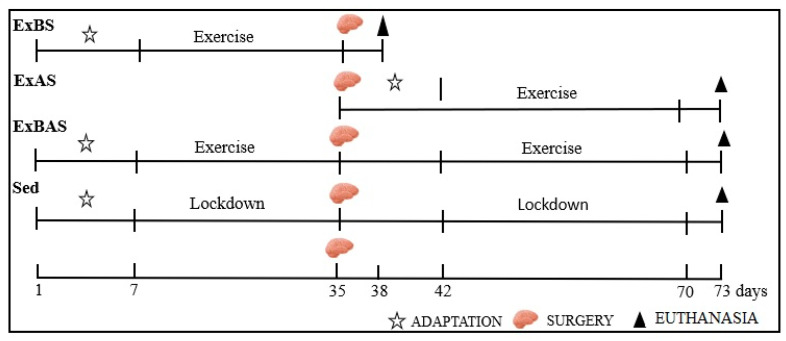
Experimental protocol: exercise before surgery (ExBS); exercise after Surgery (ExAS), exercise before and after surgery (ExBAS), and sedentary (Sed).

**Figure 2 ijerph-20-02925-f002:**
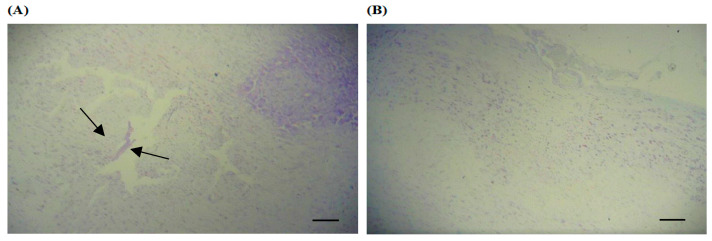
Photomicrograph of the *Substantia nigra* of the midbrain at 4× magnification. (**A**) Animal from the PD group. (**B**) Animal from the Sham Group. The arrows show the area of electrolyte damage in the substantia nigra of animals with PD.

**Figure 3 ijerph-20-02925-f003:**
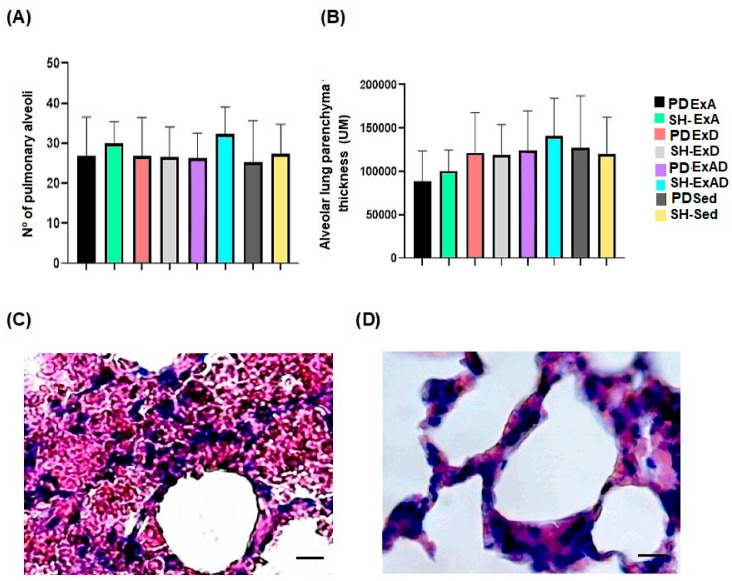
Lung Tissue Analysis. (**A**) Mean number of pulmonary alveoli. (**B**) Mean lung parenchyma thickness in the animal groups. (**C**) Photomicrographs of the lung parenchyma. (**D**) Photomicrograph of alveolar wall thickness (400× magnification).

**Figure 4 ijerph-20-02925-f004:**
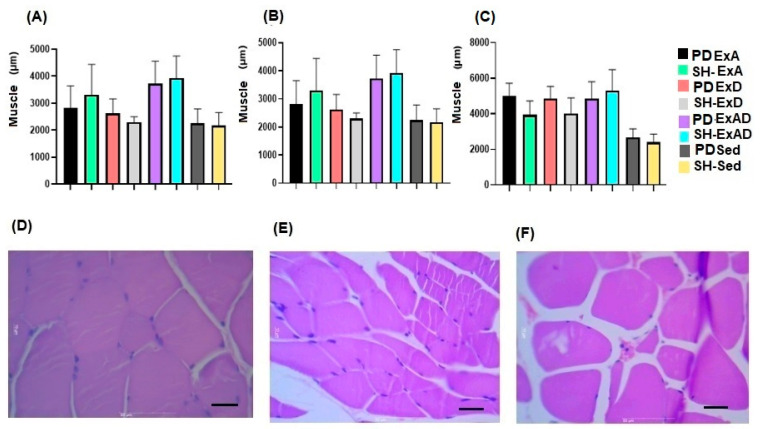
Analysis of the cross-sectional area of the respiratory muscles at 400× magnification: (**A**) Diaphragm muscle, (**B**) Intercostal muscle, and (**C**) rectus abdominis muscle. Photomicrograph of the respiratory muscles: (**D**) diaphragm muscle ExAS, (**E**) intercostal muscle ExAS, and (**F**) rectus abdominis muscle ExAS.

**Figure 5 ijerph-20-02925-f005:**
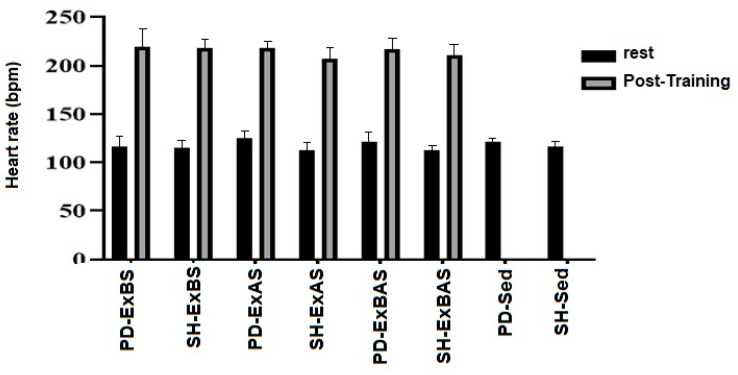
Analysis of mean heart rate at rest and after physical training among groups. The relationship between body weight and heart weight (Figure 4) showed no significant differences in the PD and SH animals belonging to the same training or sedentary protocols (*p* = 0.86).

**Figure 6 ijerph-20-02925-f006:**
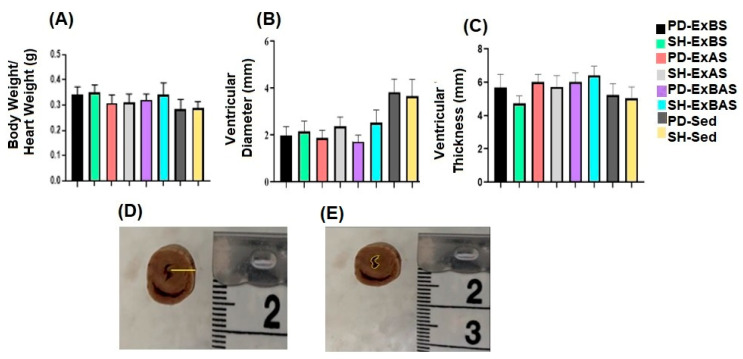
Morphometric analysis of the heart: (**A**) mean ratio of body weight to heart weight, (**B**) diameter of the left ventricle, and (**C**) thickness of the left ventricle. Photomicrograph of (**E**) diameter of the left ventricle in the PD-ExBAS group and (**D**) thickness of the left ventricle in the PD-ExBAS group.

## Data Availability

Not applicable.

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
