# Peer review of "The Effects of Resistance Exercise on the Cardiorespiratory Tissue of Rats with Parkinson’s Disease"

_ijerph, 2023, doi:10.3390/ijerph20042925_

Round 1

Reviewer 2 Report

This original research entitled "The Effects of Resistance Exercise on the Cardiorespiratory 2 Tissue of Rats with Parkinson’s Disease" focused on discussing the effects of progressive resistance exercise on cardiorespiratory tissue in Parkinson’s rats that have associated with exercise duration to explore the efficacy and safety of high-intensity progressive resistance exercise on Parkinson’s disease. For doing this purpose, Parkinson’s rats were induced by electrolyte stimulation in the substantia nigra of the midbrain. Eight group of rats (control and Parkinson) were treated 4-week resistance exercise intervention before, after, before and after surgery and the morphometric analysis were performed in the muscles of the myocardial, diaphragm, intercostal and abdominal challenge foramen using the Hematoxylin and Eosin (HE). This is an interesting study. The results are novel and of relevance. However, some major problems needed to give clear information as follows:

1. The study did show that physical exercise caused changes in heart and lung function of Parkinson’s rats. Resistance exercise to improve Parkinson's impaired cardiopulmonary function is the premise of the entire study, but this study lacks the test and analysis of cardiopulmonary function indicators. This should be added as a limitation of the study. Also, the discussion and the conclusion of the study need to be updated.

2. Animal Models: the establishment of Parkinson’s animal models is an important premise and foundation of this study, however, the presentation of the method in the ABSTRACT (not found) and the MATERIALS and METHODS (induced by electrolyte stimulation in the substantia nigra of the midbrain.) were ambiguous. Moreover, what are the criteria for successful animal model establishment?

3. The research method is too simple and single, and the evidence for muscle hypertrophy is insufficient to judge muscle hypertrophy with HE staining alone.

4. The research methods and results are too confusing to read. It is recommended to reorganize the logical relationships.

5. The bars in the histogram are similar in color, and it is not possible to confirm which group each bar corresponds to. Meanwhile, Scale bars are required for HE staining results.

6. What does Figure 3D and F depict? This result is not described in the results section of the paper. Is the number not continuous?

7. Figure 4 has only one statistical chart, what does ABCDE in the legend mean? Please clarify.

        8. Line 294-298: The paragraph is the same as the previous paragraph and is proposed for deletion.

Round 2

Reviewer 2 Report

All my concerns have been addressed reasonably.  Thank you.